# Impact of Sidestream Pre-Treatment on Ammonia Recovery by Membrane Contactors: Experimental and Economic Evaluation

**DOI:** 10.3390/membranes12121251

**Published:** 2022-12-10

**Authors:** Miguel Aguilar-Moreno, Sergi Vinardell, Mònica Reig, Xanel Vecino, César Valderrama, José Luis Cortina

**Affiliations:** 1Chemical Engineering Department, Escola d’Enginyeria de Barcelona Est (EEBE), Universitat Politècnica de Catalunya (UPC)-BarcelonaTECH, C/Eduard Maristany 10-14, Campus Diagonal-Besòs, 08930 Barcelona, Spain; 2Barcelona Research Center for Multiscale Science and Engineering, Campus Diagonal-Besòs, 08930 Barcelona, Spain; 3CETaqua, Carretera d’Esplugues, 75, 08940 Cornellà de Llobregat, Spain

**Keywords:** gas permeable membrane, coagulation–flocculation, resource recovery, circular economy, techno-economic evaluation

## Abstract

Membrane contactor is a promising technology for ammonia recovery from the anaerobic digestion centrate. However, high suspended solids and dissolved organic matter concentrations can reduce the effectiveness of the technology. In this study, coagulation–flocculation (C/F) and aeration pre-treatments were evaluated to reduce chemical oxygen demand (COD), turbidity, suspended solids and alkalinity before the ammonia recovery stage using a membrane contactor. The mass transfer coefficient (Km) and total ammonia (TAN) recovery efficiency of the membrane contactor increased from 7.80 × 10^−7^ to 1.04 × 10^−5^ m/s and from 8 to 67%, respectively, after pre-treating the real sidestream centrate. The pre-treatment results showed that dosing aluminium sulphate (Al_2_(SO_4_)_3_) at 30 mg Al/L was the best strategy for the C/F process, providing COD, turbidity and TSS removal efficiencies of 50 ± 5, 95 ± 3 and 90 ± 4%, respectively. The aeration step reduced 51 ± 6% the HCO_3_^−^ content and allowed reducing alkaline consumption by increasing the pH before the membrane contactor. The techno-economic evaluation showed that the combination of C/F, aeration and membrane contactor can be economically feasible for ammonia recovery. Overall, the results of this study demonstrate that C/F and aeration are simple and effective techniques to improve membrane contactor performance for nitrogen recovery from the anaerobic digestion centrate.

## 1. Introduction

Nutrient pollution is one of the major environmental problems due to excessive discharge of nitrogen and phosphorus into the environment. Anthropogenic activities and population growth have increased the amount of nitrogen contained in wastewater. The recovery of this nitrogen is particularly important considering that ammonia is the second most produced chemical in the world [1,2,3]. Ammoniacal nitrogen recovery has the potential (i) to reduce the dependency of the Haber–Bosch process to obtain nitrogen-based fertilizers, (ii) to produce a fertilizer (e.g., NH_4_NO_3_, (NH_4_)_2_HPO_4,_ (NH_4_)_2_SO_4_) suitable for commercialization and (iii) to reintroduce nitrogen into its cycle contributing to the circular economy [4,5]. For this reason, it is important to develop efficient technologies for nitrogen recovery to support the transition of wastewater treatment plants (WWTPs) towards water resource recovery facilities (WRRF) [6].

Several technologies have been proposed to recover nitrogen from wastewater treatment plants (WWTPs), such as ion exchange (IX) technologies [7], membrane contactors (MC) [8,9,10] or ultrafiltration (UF) [11]. For instance, Wan et al. [12] effectively recovered nutrients from the sludge fermentation liquor in a WWTP (N-NH_4_^+^ and P-PO_4_^3-^) using natural zeolites and proposed a model to predict that a maximum recovery of 94% ammonium and 98% phosphate could be achieved. Among them, ammoniacal nitrogen recovery through membrane contactors has been reported as a suitable technology to achieve high nitrogen recovery efficiencies with relatively low energy inputs [5]. By this technology, ammonia in gas form diffuses through a porous hydrophobic membrane from the feed solution to the acidic stripping solution. Subsequently, it can be recovered in ammonium form as a nitrogen-rich fertilizer. [13]. Vecino et al. [14] used a membrane contactor for ammonium recovery as a nutrient-based fertilizer product and achieved a maximum ammonium recovery of 94% using a regenerated stream with ion exchange from an initial sidestream wastewater. Sheikh et al. [15] also achieved similar values (>95%) of recovery using synthetic water and liquid–liquid hollow fibre MC (LL-HFMC). Additionally, both membrane contactors and ion exchange technologies can be combined as proposed by Sancho et al. [16]. In that study, a concentrated ammonium stream was generated by means of liquid–liquid membrane contactors, by previously passing it through zeolites, achieving a recovery of 95% [16]. Thus, these publications highlight that membrane contactors have potential to achieve high recovery efficiencies and to obtain ammonium-free streams.

However, membrane contactors still need to overcome some challenges when using streams with high concentration of organic matter. Membrane fouling, caused by organic matter and/or suspended solids, can lead to the deposition of solids as a thin cake layer and increase pore clogging [17]. This phenomenon generates a reduction in the flux during long-term operation. Thus, to maintain adequate flux levels, it is necessary to increase energy and chemical consumption with a direct impact on the membrane lifetime and economic feasibility [18]. In this regard, some pre-treatment strategies have been proposed to reduce fouling of membrane contactors, such as UF [19], coagulation–flocculation (C/F) processes [17] or ion exchange [20]. For example, Rivadeneyra et al. [20] used ion exchange technology and observed a maximum chemical oxygen demand (COD) removal efficiency of 70% with an initial COD load of 4500 mg O_2_/L. Raghu et al. [21] combined ion exchange with coagulation–flocculation and achieved a COD removal of 80% from an industrial wastewater effluent.

C/F consists of destabilization of colloids by surface modification. This reduces the electrostatic repulsive forces between the particles and leads to the formation of larger flocs with improved settling properties [22]. The most common coagulants and flocculants used are iron and aluminium salts because these chemicals have demonstrated their effectiveness to reduce the chemical oxygen demand (COD) of liquid streams [23,24]. C/F has been widely applied in wastewater treatment applications as it allows removing organic and inorganic matter with relatively low costs [19,25]. For instance, Al-Juboori et al. [26] evaluated the use of PAX/polymer or starch as a coagulant to pre-treat the centrate before a membrane contactor. 

Besides C/F, aeration could also be a useful pre-treatment to reduce the amount of chemicals needed to increase the pH before the membrane contactor stage. Garcia-Gonzalez et al. [27] applied low flow-rate aeration and increased the pH above 8.5 before the membrane contactor, which allowed reducing the operating costs of ammonia recovery by 57%. However, to the best of the authors’ knowledge, the combination of C/F technology with aeration has not yet been used to pre-treat anaerobic digester centrate prior to a membrane contactor. Therefore, an experimental and economic study is needed to understand how C/F pre-treatment impacts the technical and economic competitiveness of implementing a membrane contactor system for nitrogen recovery.

The aim of this work is to evaluate the combination of C/F, aeration and membrane contactor processes to recover ammoniacal nitrogen from the effluent of an anaerobic digester (centrate). To this end, different operating conditions and chemical reagents were evaluated for the C/F process. After the C/F process, an aeration stage was used to reduce the amount of bicarbonates in the centrate with a direct impact on the amount of chemicals needed for pH adjustment. Subsequently, the pre-treated centrate was fed to a membrane contactor system to understand how pre-treatment conditions impacted the performance of the membrane contactor and ammonium recovery efficiency. Finally, the economic potential of implementing these pre-treatment technologies before the membrane contactor was analyzed.

## 2. Materials and Methods

### 2.1. Chemical Reagent and Wastewater Source

Three types of coagulants were used for the coagulation–flocculation tests: (i) aluminium sulphate (Al_2_(SO_4_)_3_ 18·H_2_O) from Panreac^®^ with a 96% of purity, (ii) iron chloride (FeCl_3_) from Acros Organics^®^ with a 98% of purity and (iii) a commercial coagulant HT20 from Derypol^®^. On the other hand, a mixture of Magnetite (Fe_3_O_4_) from Aldrich^®^ with a 98% purity and silicon oxide (SiO_2_) from Merck^®^ with a purity of 98% (relation of 30:70%) was used as flocculant. 

Different reagents were used for the chromatographic analysis: Methanesulfonic acid (CH_3_SO_3_H, 99%), sodium hydrogen carbonate (NaHCO_3_, 99%), anhydrous sodium carbonate (Na_2_CO_3_, 99%), nitric acid (HNO_3_ 69%) and sodium hydroxide (NaOH 1 M). All these chemicals were analytical grade reagents and were supplied by Sigma-Aldrich.

The wastewater used in this study was the anaerobic digester centrate from a municipal WWTP located in the region of Barcelona (Spain). The centrate was decanted before the tests for 24 h to reduce its concentration of COD, total suspended solids (TSS) and turbidity. The centrate used for the C/F tests contained COD and total ammonia nitrogen (TAN) concentrations of 786 mg COD/L and 650 mg N/L, respectively, which were within the range reported in the literature [28,29]. It is worth mentioning that the water used for the flocculant tests came from the same location and had a similar ion concentration to that used in the other tests, although it contained a higher COD concentration (1650 mg COD/L).

### 2.2. Experimental Design

The study was divided into 2 distinct stages (Figure 1). The first stage corresponded to the pre-treatment stage, selection of the optimum coagulant reagent and setting the optimum operating conditions with a specialized experimental design program. The specialized software allowed optimization of the mixing speed, mixing time and sedimentation time to maximize COD, TSS and turbidity removal efficiencies. Besides C/F, an aeration column for the removal of carbonate and the consequent increase in the pH was also considered. In the second stage, the performance of the membrane contactor (pH, concentration factor, ammonium recovery percentage) was tested with the untreated sidestream water and with the pre-treated water to evaluate the effectiveness of the pre-treatment on membrane contactor performance. Finally, an economic analysis was conducted to evaluate the feasibility of the application of this process train.

### 2.3. Experimental Set-Up

#### 2.3.1. Coagulant Selection

The selection of the best coagulant reagent and dosage was based on combining literature screening and lab-scale tests. Initial bibliographic research was carried out to determine the most common coagulants (Table 1) and it was observed that the most widely used coagulants were based on metals, such as aluminium or iron. After this initial screening, aluminium sulphate (Al_2_(SO_4_)_3_), iron chloride (FeCl_3_) and a commercial coagulant Derypol^®^ HT20 (which is in the category of vegetable coagulants) were chosen.

The lab-scale tests were conducted in a Jar-test set-up (Jar-test *OVAN*^®^
*JT60 E*), which consists of (i) six rotating stirring rods with adjustable speed and height and (ii) six beakers filled with 500 mL of the centrate under study. Two set of experiments were conducted to determine the best coagulant and the dosage strategy for the C/F process.

The first set of experiments was designed to determine the two most favourable coagulants. In these tests, the type of coagulant was changed, while keeping the operating conditions constant. The dosage was set at 50 mg/L and the mixing time was 5 min at a mixing speed of 200 rpm (see Appendix A), which was based on available literature [31,34,35,36]. The experiments were conducted in triplicate. The second set of experiments was designed (i) to determine the optimum dosage for the two flocculants selected in the previous experiments and (ii) to obtain the most favourable coagulant at this optimum dosage. All the coagulant dosages referred to the quantity of metal added.

The impact of dosage on the efficiency of the C/F process was evaluated for the best coagulant. To this end, the dosage was varied from 10 to 800 mg/L with the Jar-test conditions mentioned above. Table 2 lists the experimental conditions for these tests. The experiments were conducted in triplicate.

#### 2.3.2. Determination of the Optimal Operational Conditions for the C/F Process

Once the optimum coagulant chemical and dosage were selected, the most favourable operational parameters (i.e., mixing time, mixing speed and settling time) were determined by using the Jar-test equipment. For this purpose, a design program was used to optimize the number of tests required and to determine the best operational conditions for the C/F process.

The Design Expert^®^ 11 software was used following the factorial design of Box–Behnken, which is based on dependent and independent variables [37]. The dependent variables were those investigated and measured in the study, whereas the independent variables were modified to study their effect on the dependent variables [38]. Table 3 summarizes the dependent variables studied in this work. The coded variables were assigned values of +1 (maximum), 0 (central) and −1 (minimum) depending on the variation of each variable.

The Box–Behnken design is a rotating or quasi-rotating second-order experimental design based on incomplete three-level factorial designs. The number of experiments (N) needed according to the Box–Behnken design can be obtained from Equation (1).
(1)$$N=2\cdot k\left(k\;-\;1\right){+C}_{0}$$
where k is the number of variables, and $C_{0}$ is the number of central points [8,39]. In this case, three variables (MT, MV and RT) and five central points were studied resulting in seventeen experiments. The Box–Behnken experimental designs were applied by means of Equations (3) and (4) [8].
(2)$${y=\beta }_{0}\overset{k}{\underset{i=1}{\sum }}\beta_{i}X_{i}\overset{k}{\underset{i=1}{\sum }}\sum_{j\geq 1}^{k}\beta_{\mathrm{ij}}X_{i}X_{j}+\varepsilon$$
(3)$${y=\beta }_{0}\overset{k}{\underset{i=1}{\sum }}\beta_{i}X_{i}{+\beta }_{0}\overset{k}{\underset{i=1}{\sum }}\beta_{\mathrm{ii}}X_{i}^{2}+\overset{k}{\underset{i=1}{\sum }}\sum_{j\geq 1}^{k}\beta_{\mathrm{ij}}X_{i}X_{j}+\varepsilon$$
where β_o_ is the constant factor, β_i_ represents the coefficients of the linear parameters, k is the number of variables, X_i_ and X_j_ represent the independent variables, ε is the residual factor associated with the experiments, y is the dependent variable, β_ij_ represents the coefficients of the interaction parameters and β_ii_ represents coefficients of the quadratic values.

Finally, the software allows for analysis of the obtained results to provide the optimal conditions (e.g., removal of each of COD, TSS and turbidity) through the analysis of graphics and data.

#### 2.3.3. Coagulation Test for the Optimal Coagulant Conditions and Dosage

The optimal coagulant and dosage obtained from stage 1 and 2 were tested to determine the experimental COD, TSS and turbidity removal efficiencies under the most favourable conditions. In this assay, the optimal conditions determined by the two previous tests were applied in the Jar-test equipment and it was verified if the theoretical results provided by the experimental design software were experimentally fulfilled. The experiments were conducted in triplicate.

#### 2.3.4. Flocculation Test

Flocculation tests were conducted to evaluate if combining coagulant and flocculant addition improves solids removal efficiency when compared with stand-alone coagulant addition. The flocculation experiments were carried out with the optimal conditions obtained from the previous experiments and adding different dosages (0–50 mg/L) of a clay-based flocculant (Fe_3_O_4_(s) and SiO_2_(s)) that works effectively with metal-based coagulants for COD reduction [33]. The flocculant was prepared by pulverizing and mixing Fe_3_O_4_ and SiO_2_ with a relation of 30% Fe_3_O_4_ and 70% of SiO_2_. Appendix A shows the operational parameters used for the flocculation tests.

#### 2.3.5. Aeration Tests

The possibility of adding an aeration stage [27,40,41] was evaluated: (i) to increase the pH of the centrate and (ii) to reduce the concentration of carbonates present in the sample. The aeration tests were carried out in an open aeration column of 3.5 m height and 30 cm diameter with a capacity of 25 L. The air was introduced at the bottom of the tank through an electric compressor at a flow rate of 2 Nm^3^/h. The column was filled with the centrate and a constant air flow rate (364 L/h) was applied for a period of time adequate to cause reactions described by Equations (4)–(6).
(4)$${\mathrm{HCO}}_{3(\mathrm{aq})}^{-}{+H}_{(\mathrm{aq})\;}^{+}↔{\;\mathrm{CO}}_{2(\mathrm{aq})}{+H}_{2}O_{(l)\;}$$
(5)$${\mathrm{CO}}_{2(\mathrm{aq})}\;↔{\;\mathrm{CO}}_{2(g)}$$
(6)$${\mathrm{NH}}_{4(\mathrm{aq})}^{+}\;↔{\;\mathrm{NH}}_{3(g)\;}{+H}_{(g)\;}^{+}$$

Thus, these experiments allowed bicarbonate conversion to CO_2_(g) (aq) /Equation (4) due to the aeration process promoting the removal of dissolved CO_2_(g) (aq) as CO_2_(g) (Equation (5)) and consequently increasing the pH. Subsequently, the pH increased allowed the conversion of NH_4_^+^ into ammonia. (Equation (6)). The aeration experiments were performed in duplicate.

#### 2.3.6. Flat-Sheet Membrane Contactor

The different pre-treatment processes were aimed at conditioning the centrate to reduce fouling and clogging in the membrane contactor. A flat-sheet membrane contactor similar to the one used by Hasanoĝlu et al. [10] was used in this study. The polytetrafluoroethylene (PTFE) membrane had a surface area of 90 cm^2^ and a pore size of 0.2 µm. The pH of the feed solution was increased up to 10.2 with NaOH 1 M, to displace the equilibrium towards NH_3_. The feed solution was stored in a 5 L tank, whereas the acid stripping solution (0.4 M nitric acid) was stored in a 1.5 L tank. Both tanks were continuously agitated, while nitric acid was continuously added to maintain the pH of the stripping solution in the acidic regime (pH < 2). The feed and stripping solutions were circulated at 450 mL/min in counter current mode towards both sides of the membrane. Further details of the membrane contactor set-up can be found elsewhere [9]. 

The ammonia flux through the membrane is driven by the difference between the partial pressure on both sides of the membrane, ($p_{{\mathrm{NH}}_{3,}f}\;-\;p_{{\mathrm{NH}}_{3},s}$) and the mass transfer coefficient $\left(K_{m\left({\mathrm{NH}}_{3}\right)}\right)$ (Equation (7)).
(7)$$J_{{\mathrm{NH}}_{3}}=\frac{K_{m\left({\mathrm{NH}}_{3}\right)}{(p}_{{\mathrm{NH}}_{3,}f}\;-\;p_{{\mathrm{NH}}_{3},s})}{\mathrm{RT}}$$
where $p_{{\mathrm{NH}}_{3},s}$ is the partial pressure of ammonia in the shell side (atm), $p_{{\mathrm{NH}}_{3,}f}$ is the partial pressure in the feed side (atm),$K_{m\left({\mathrm{NH}}_{3}\right)}$ is the ammonia mass transfer coefficient (m/s), R is the universal gas constant coefficient (0.082 atm·m^3^/k mol·K) and T is the temperature of the system (K).

Subsequently, Equation (7) can be expressed as Equation (8) considering that: (i) the partial pressure of ammonia on both sides of the membrane can be assumed as the concentration of ammonia on either side, (ii) the pH does not vary during the experimental procedure, meaning that the concentration of ammonia is proportional to the TAN concentration in the feed solution and (iii) the ammonia partial pressure in the stripping side is negligible [8,14].
(8)$$\ln\frac{C_{0\left({\mathrm{NH}}_{3}\right)f}}{C_{t\left({\mathrm{NH}}_{3}\right)f}}=\frac{K_{m\left({\mathrm{NH}}_{3}\right)}A_{m}}{V_{f}}t$$
where $A_{m}$ is the membrane area (m^2^), $C_{0\left({\mathrm{NH}}_{3}\right)f}$ and $C_{t\left({\mathrm{NH}}_{3}\right)f}$ are the feed ammonia concentration (mg/L) at the initial time and at the experimental time, respectively, and $V_{f}$ is the feed volume (m^3^).

The tests were conducted for both untreated and pre-treated centrate to evaluate and compare the membrane contactor performance before and after pre-treatment implementation.

### 2.4. Analytical Methods

The anions and cations were analyzed by an ion chromatography system (Dionex ICS-1000 and ICS-1100 Thermo-Fisher Scientific, USA) equipped with a cationic detector (ICS-1000) and an anionic detector (ICS-1100) and controlled by Chromeleon^®^ chromatographic software. A CS16 column (4 × 250 mm) and an AS23 column (4 × 250 mm) (Phenomenex, Barcelona, Spain) were used for cation and anion determination and quantification, respectively. The mobile phase was a 0.03 mol/L CH_3_SO_3_H solution for the cation system, and a mixture of 0.8 mmol/L NaHCO_3_ and 4.5 mmol/L Na_2_CO_3_ for the anion system.

The COD was analyzed through the Standard Method 5220C using a multiparametric photometer HI83224 (Hanna Instruments, Padua, Italy), whereas TSS were analyzed through the Standard Method 2540D [42]. A turbidimeter HI 93703 (Hanna instruments, Padua, Italy) was used to measure the turbidity. Total alkalinity was measured by titration following the Standard Method 2320B and using a T70 titrator (Mettler Toledo, Columbus, OH, United States).

### 2.5. Economic Analysis

An economic analysis was conducted to evaluate the techno-economic implications of implementing a membrane contactor system for ammonia recovery from the anaerobic digester centrate. Appendix A shows the configuration evaluated in the economic analysis, which included four different stages: (i) C/F with Al_2_(SO_4_)_3_ to enhance solids sedimentation, (ii) precipitation for suspended solids removal, (iii) aeration to desorb part of the solubilized CO_2_ and reduce the alkalinity and (iv) membrane contactor system for nitrogen recovery. The membrane contactor system was operated by using an HNO_3_ trapping solution and considering a relation between the feed and trapping solution flow rate of 1:1. The pH of the feed solution was adjusted to 10.2 with NaOH to displace the NH_4_^+^/NH_3_ equilibrium towards NH_3_. The trapping solution was continuously recirculated from the acid tank to the membrane contactor and replaced when the pH increased by up to 6 [28]. The mass balance was obtained considering that the WWTP generated 150 m^3^/day of centrate, containing TAN and TSS concentrations of 0.71 g N/L and 0.24 g TSS/L, respectively. Detailed information on the mass balance can be found in Appendix A.

The capital costs, operating costs and revenues were calculated using both lab-scale data and literature average values. The capital costs accounted for membrane contactor, tanks, stirrers, blowers and pumps, whereas the operating costs accounted for energy consumption, sludge disposal, equipment replacement and the purchase of chemicals (i.e., Al_2_(SO_4_)_3_, NaOH and HNO_3_). Finally, the revenues were obtained considering (i) the commercialization of the produced NH_4_NO_3_ and (ii) the lower nitrogen load to be treated in the mainstream of the WWTP. Appendix A summarize the main design and cost parameters used for the economic analysis. 

The present value (PV) of the gross cost and revenues was calculated for the nitrogen recovery configuration by using Equations (9) and (10), respectively. Subsequently, Equation (11) was used to calculate the net present value (NPV):(9)$${\mathrm{PV}}_{\mathrm{GC}}=\mathrm{CAPEX}+\overset{T}{\underset{t=1}{\sum }}\frac{{\mathrm{OPEX}}_{t}}{{\left(1+i\right)}^{t}}$$
(10)$${\mathrm{PV}}_{R}=\overset{T}{\underset{t=1}{\sum }}\frac{R_{t}}{{\left(1+i\right)}^{t}}$$
(11)$$\mathrm{NPV}=\overset{T}{\underset{t=1}{\sum }}\frac{R_{t}{\;-\;\mathrm{OPEX}}_{t}}{{\left(1+i\right)}^{t}}\;-\;\mathrm{CAPEX}$$
where CAPEX is the capital expenditure (EUR), OPEX_t_ is the operating expenditure at year t (EUR), R_t_ is the revenue at year t (EUR), PV_GC_ is the PV of the gross cost (EUR), PV_R_ is the PV of the revenues (EUR), NPV is the net present value (EUR), i is the discount rate (5%) and T is the plant lifetime (20 years).

## 3. Results and Discussion

The following sections discuss the results concerning the application of C/F and aeration pre-treatments before a membrane contactor. Table 4 shows the COD, TSS, turbidity and ion concentrations of the centrate wastewater used for these tests.

### 3.1. Coagulant and Dosage Selection for the C/F Process

Table 5 collects the COD and turbidity removal efficiencies for the three coagulants (FeCl_3_, Al_2_(SO_4_)_3_ and Derypol^®^ HT20) analyzed in this study. Al_2_(SO_4_)_3_ reported the best COD removal efficiencies (50.2 ± 1.1%), followed by FeCl_3_ (38.9 ± 0.3%) and Derypol HT20 (36.0 ± 0.3%). Thus, Al_2_(SO_4_)_3_ and FeCl_3_ were selected for the next set of experiments. The turbidity removal efficiencies ranged from 74.2 to 84.7%. The lowest turbidity values were obtained by using FeCl_3_ (74.2 mg/L) and they were similar to those achieved by Abdessemed et al. [43], which achieved turbidity removal values of 66.1% using FeCl_3_.

Table 6 lists the COD and turbidity removal efficiencies of Al_2_(SO_4_)_3_ and FeCl_3_ for concentrations ranging from 10–800 mg/L. The results showed that Al_2_(SO_4_)_3_ provided better COD removal performance in comparison to FeCl_3_, which reinforces the idea that Al_2_(SO_4_)_3_ is the most favourable coagulant–flocculant to be used as a membrane contactor pre-treatment. On the one hand, the COD removal efficiency increased from 42.5 to 51.8% as the FeCl_3_ concentration increased from 10 to 800 mg/L, respectively. On the other hand, the COD removal efficiency increased from 51.5 to 62.1% as the Al_2_(SO_4_)_3_ concentration increased from 10 to 200 mg/L, respectively. However, in the case of Al_2_(SO_4_)_3_, dosages above 200 mg/L only led to minimal improvements in the COD removal efficiency. This behaviour is due to the fact that applying coagulant dosages above the optimal level does not lead to considerable improvements [44].

The results also showed that the pH progressively decreased as the coagulant dosage increased. In the case ofAl_2_(SO_4_)_3_, when the metal ion (Al^+3^) hydrolyzes in water, it reacts to form complex (Al(OH)_n_ ^+(n−3)^) compounds. This leads to the formation of CO_2_(g), which increases the acidity of the solution [23]. From the results of Table 6, it can be concluded that dosing 30 mg/L of Al (Al_2_(SO_4_)_3_) can be considered as the optimum strategy because this dosage achieved similar COD removal efficiencies than those achieved above 200 mg/L, while reducing the coagulant dosage more than seven times.

### 3.2. Optimization of the Operating Conditions for the C/F Process

After selecting the optimum coagulant and dosage (Al_2_(SO_4_)_3_, 30 mg Al/L), the impact of the operational conditions (i.e., mixing time, mixing speed and settling time) on the C/F efficiency was evaluated. Seventeen experiments were tested based on the outputs provided by the Design Expert 11 software (see Appendix A for further details on the experimental conditions tested). These experiments were conducted changing the mixing time, the mixing speed and the settling time. Figure 2 shows the theoretical TSS, turbidity and COD removal values obtained from the Design Expert 11 software for the different mixing time and mixing speed conditions at a fixed settling time of 30 min. It is worth mentioning that only the results of 30 min settling time are illustrated because this condition provided the best results when compared with the other settling times. The results highlighted that reducing the mixing time to 5 min and the mixing speed to 100 rpm, would theoretically increase removal values up to 100% in turbidity and suspended solids and up to 70% in COD. Accordingly, the software revealed that there was better removal when mixing time and speed were reduced to the minimum tested values. This behaviour was in agreement with Kan et al. [45], who reported that higher mixing speed did not give a better coagulation performance.

Subsequently, coagulation tests were carried out with the optimum conditions obtained from the software. Table 7 illustrates the results of these tests in terms of TSS, turbidity and COD removal values.

The removal values showed an improvement compared with the previous test (58.1 ± 0.3 COD, 94.9 ± 0.2 TSS and 89.8 ± 0.8 turbidity), although the values predicted by the design software were not achieved. Guimarães et al. [46] tested several coagulants (including aluminium sulphate at 40 mg/L Al) and reached COD removal efficiencies (38%) below those achieved in this study (58%). On the other hand, Salem et al. [47] reported turbidity removal efficiencies of 86%, which were similar than those achieved in this study (90%).

### 3.3. Flocculation Stage

Figure 3 shows the obtained values of COD and turbidity removal for the different dosages of flocculant Fe_3_O_4_/SiO_2_ (30–70% (*w*/*w*)) added. A test without flocculant was also conducted, which consisted of applying the optimum dosages and parameters obtained from the coagulant stage tests (Section 3.1). The results illustrated maximum COD removal (89.7%) when the flocculant dosage was 10 mg/L and maximum turbidity removal (83.6%) when the dosage was increased up to 30 mg/L. In all the tests, the TSS removal values remained practically constant around 95%. Sultana et al. [48] treated wastewater with an organic concentration (745 mg O_2_/L) similar to the present study water (786 mg O_2_/L) using aluminium sulphate coagulant and clay-based flocculant. The authors obtained COD removal efficiencies of 46.7%, which are below those achieved in this study. On the other hand, Preston et al. [49] worked with wastewater with a similar turbidity (300 NTU) than that of the present study (275 NTU), using aluminium sulphate as coagulant and Moringa as natural flocculant, and reached a similar turbidity removal of 96.2%. Overall, Figure 3 results revealed that the addition of Fe_3_O_4_(s)/SiO_2_(s) only led to small improvements concerning removal values.

According to the results obtained, it could be concluded that the addition of coagulant + flocculant did not provide a consistent positive improvement compared to the addition of only coagulant.

### 3.4. Aeration Stage

An aeration step was added after coagulation–flocculation to promote CO_2_(g) stripping to reduce alkalinity and increase the pH before the membrane contactor system [41]. Figure 4 shows the evolution of HCO_3_^−^ removal and pH over the aeration time. The HCO_3_^−^ present in the centrate was reduced by about 50% after 240 min of constant aeration, although almost 30% of elimination was reached after 15 min. The results showed that after 1 h of operation time, a compromise between carbonates removal (34%) and pH increase (8.83) was achieved, although higher removal values could be reached at expenses of higher times of operation. This agrees with the pH results, where a sudden increase was observed after 15 min of aeration, reaching a constant value after 240 min. It is worth mentioning that the application of aeration could also lead to NH_3_ losses due to volatilization, although they did not account for more than 2% in our study (data not shown).

García-González et al. [27] also used an aeration system as a membrane contactor pre-treatment stage. The aeration system increased the pH above 8.5, which allowed the partial displacement of NH_4_^+^/NH_3_ equilibrium towards NH_3_ without the addition of external chemicals. Besides technical aspects, aeration implementation has the potential to reduce the total cost of the process by 70% due to the reduction in alkaline purchasing cost (Dube et al., 2016). It is also relevant to mention that it is possible to use recycled chemicals to further reduce the operating cost of the system. 

### 3.5. Flat-Sheet Membrane Contactor Stage

Figure 5 shows the membrane contactor results for the treated and untreated centrate during the experimental time. The results illustrated that the TAN recovery efficiency increased from 7.5 to 66.6% after implementing the pre-treatment train (Figure 5B). This highlighted that C/F and aeration pre-treatments are crucial to improve the TAN recovery efficiency from the anaerobic digester centrate using membrane contactors. In the case of the pre-treated centrate, the TAN concentration in the feed solution decreased from 0.9 g/L to 0.3 g/L (Figure 5A), whereas the TAN concentration in the acid solution increased from 0 to 2.7 g/L (Figure 5C). This agrees with the outputs of other studies recovering TAN using membrane contactors [10,14]. Similarly, the results obtained in terms of concentration factor are in line with the results of TAN in the acid tank. The concentration factor corresponded to 3.8 and was obtained from the relationship between the ammonium concentration in the acid tank (3.5 g/L) and the initial ammonium concentration in the feed tank (0.9 g/L).

Besides the TAN recovery efficiency, the ammonia mass transfer coefficient (K_m_) was also calculated. The K_m_ of the pre-treated centrate (1.04 × 10^−5^ m/s) was almost two orders of magnitude higher than that achieved with the non-treated centrate (7.80 × 10^−7^ m/s). These results corroborate that the implementation of C/F and aeration before the membrane contactor is needed to achieve efficient TAN recoveries from the anaerobic digester centrate. Interestingly, the K_m_ achieved in the present study with the pre-treated centrate and flat-sheet membrane contactors was higher in comparison with K_m_ values reported in the literature using hollow fibre contactors (Table 8). The highest Km achieved in this study could be attributed to the high efficiency of the pre-treatment process since COD, TSS and turbidity were substantially reduced. This led to almost negligible fouling, no clogging and no reduction in ammonia transfer during the operation of the membrane contactor for the pre-treated centrate.

The results of this study clearly confirmed that, in the case of a centrate with a high concentration of organic matter and suspended solids, pre-treatment using C/F and aeration can improve the performance of the membrane contactor. The pre-treatment application allows avoiding operating problems, such as loss of hydrophobicity due to biofouling and clogging of the membrane, improving the membrane recovery performance and making it technically feasible.

### 3.6. Economic Analysis

#### 3.6.1. Economic Feasibility of Membrane Contactor Implementation

Figure 6 illustrates the economic balance of implementing a membrane contactor system to recover ammonia from the anaerobic digester centrate. The results show that membrane contactor implementation in a WWTP led to a negative NPV. Ammoniacal nitrogen recovery from the anaerobic digester centrate allows (i) achieving revenues from the ammonium nitrate fertilizer produced and (ii) reducing the nitrogen load to the mainstream of the WWTP with a direct impact on energy consumption. However, these revenues did not offset the additional costs associated with the construction and operation of the different process units. From these results, it is conceivable to state that further improvements are still necessary to make nitrogen recovery through membrane contactors economically attractive. Besides economic considerations, ammoniacal nitrogen recovery from the anaerobic digester centrate has the potential to reduce disturbances in the mainstream nitrification–denitrification process and improve the WWTP effluent quality [52,53].

The membrane contactor system was the costliest process (55%), followed by aeration (36%) and coagulation–flocculation (9%) (see Appendix A). The high cost of the membrane contactor system is mainly associated with the intensive consumption of HNO_3_ and, to a lesser extent, NaOH. In this regard, chemical consumption features the highest cost contribution, representing 57% of the gross cost (Appendix A). Energy consumption also represents an important fraction of the gross cost (34.1%), which can be attributed to the high energy requirements of the air blower system. These results highlight that chemical consumption and aeration requirements are two important operational factors influencing the economic competitiveness of the system.

#### 3.6.2. Sensitivity Analysis

Figure 7 shows the sensitivity analysis for a ± 30% variation of the main economic parameters. The results illustrate that the NH_4_NO_3_ price featured the highest impact on the NPV. This is particularly important considering that the cost of fertilizers is expected to increase in the future due to the progressive increase in fuel and electricity costs [54]. To better understand how NH_4_NO_3_ price impacts the economic balance of the system, a sensitivity analysis was conducted for NH_4_NO_3_ prices between 0.30 and 0.70 EUR/kg (Figure 8). The results show that the NPV of ammoniacal nitrogen recovery increased from EUR −350,000 to 300,000 as the NH_4_NO_3_ price increased from 0.30 to 0.70 EUR/kg, respectively. This implies that a positive NPV was achieved at NH_4_NO_3_ prices above 0.52 EUR/kg. Overall, these results highlight that the commercialization of the produced NH_4_NO_3_ fertilizer has the potential to make membrane contactor configuration economically feasible.

Nitric acid and electricity costs also feature a noticeable impact on the NPV of the system (Figure 7). This reinforces the idea that chemical consumption and aeration requirements are two important aspects influencing the economics of this configuration. Conversely, membrane purchase cost variation did not lead to important changes in the NPV. The low impact of the membrane purchase cost on NPV can be attributed to the high K_m_ coefficient (1.04 × 10^−5^ m/s) achieved in this study, which is substantially higher than in other studies [5,55]. However, it is worth mentioning that the K_m_ could be substantially lower during long-term membrane contactor operation due to organic and inorganic membrane fouling development on the membrane surface. For this reason, a sensitivity analysis was conducted to evaluate the impact of K_m_ on the economic balance of the nitrogen recovery scheme under study (Figure 8).

The results show that the NPV slightly decreased from EUR −140,000 to −260,000 as the K_m_ decreased from 1 × 10^−5^ to 1 × 10^−6^ m/s, respectively (Figure 8). However, a sharp decrease in the NPV was observed at K_m_ values below 1 × 10^−6^ m/s. These results highlight that K_m_ could have a large influence on the economic balance due to its impact on the membrane requirements of the system. For this reason, it is important to look for suitable physical and chemical cleaning strategies able to achieve effective control of long-term membrane fouling without excessive consumption of chemicals and energy.

## 4. Conclusions

This study evaluated the implementation of C/F and aeration pre-treatments prior to a membrane contactor stage to recover nitrogen from the anaerobic digester centrate. The results revealed that dosing aluminium sulphate at 30 mg Al/L was the best strategy for the coagulation process. The maximum COD, turbidity and TSS removals (58 and 95 and 90%, respectively) were achieved with a mixing speed of 100 rpm, a mixing time of 5 min and a settling time of 30 min. The flocculation stage using Fe_3_O_4_(s)/SiO_2_(s) (30–70% (*w*/*w*)) did not lead to noticeable improvements in the removal efficiencies. The aeration stage reduced HCO_3_^−^ content up to 51% and increased the pH up to 9, without the addition of external chemicals. Subsequently, the effluent from the C/F and aeration stages was fed to the membrane contactor for nitrogen recovery. The membrane contactor recovered 67% of TAN and achieved a concentration factor in the acid solution of 3.8. Finally, the techno-economic evaluation showed that the combination of C/F, aeration and membrane contactor has the potential to be an economically competitive alternative for nitrogen recovery.

## Figures and Tables

**Figure 1 membranes-12-01251-f001:**
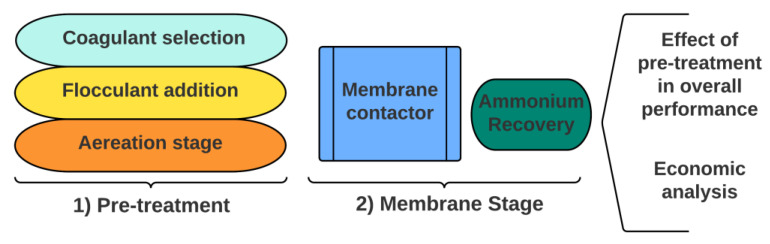
General scheme of the different anaerobic sidestream treatment stages used in the present study.

**Figure 2 membranes-12-01251-f002:**
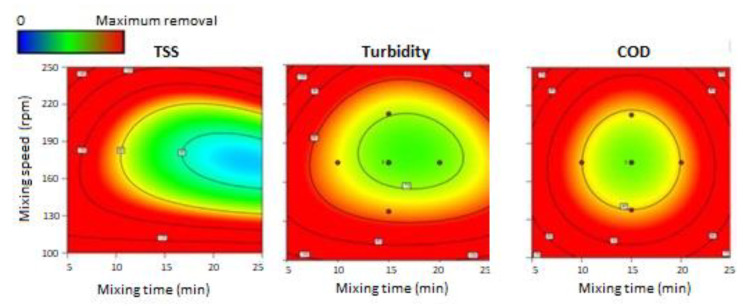
Theoretical TSS, turbidity and COD removal values for different mixing times and mixing speeds, at a fixed settling time of 30 min (graphics obtained from the Design Expert 11 software).

**Figure 3 membranes-12-01251-f003:**
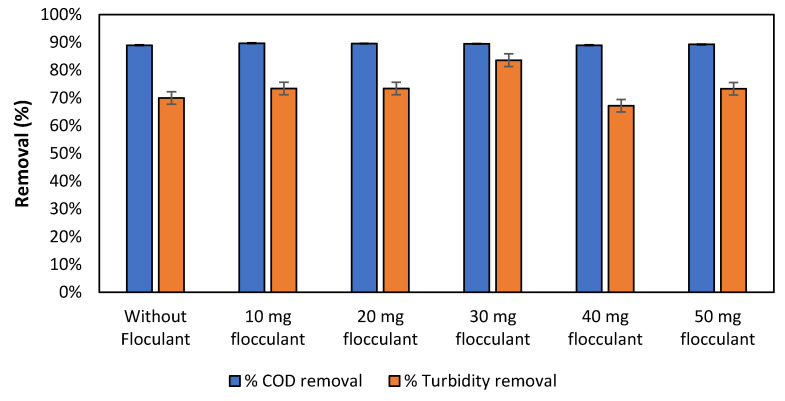
Removal of COD (%) and turbidity (%) from anaerobic centrate after Fe_3_O_4_(s)/SiO_2_(s) addition.

**Figure 4 membranes-12-01251-f004:**
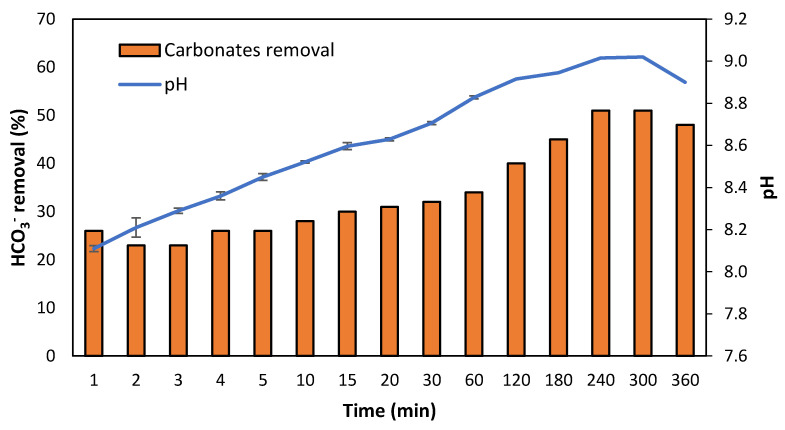
Variation of pH and the efficiency of HCO_3_^−^ removal with time in the aeration stage.

**Figure 5 membranes-12-01251-f005:**
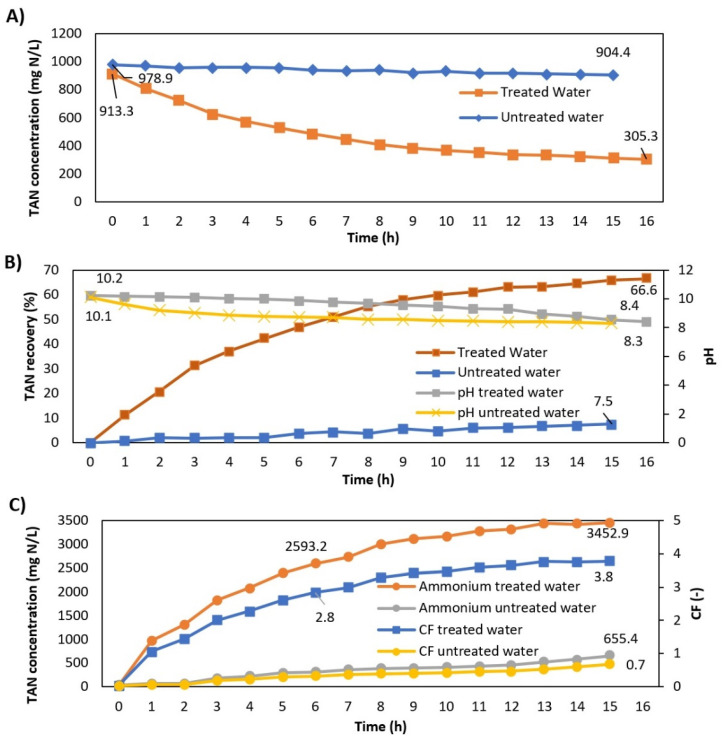
Membrane contactor results during operation: (**A**) TAN concentration evolution in the feed tank for pre-treated and untreated centrate, (**B**) TAN recovery and pH variation and (**C**) TAN concentration evolution and concentration factor in the acid tank.

**Figure 6 membranes-12-01251-f006:**
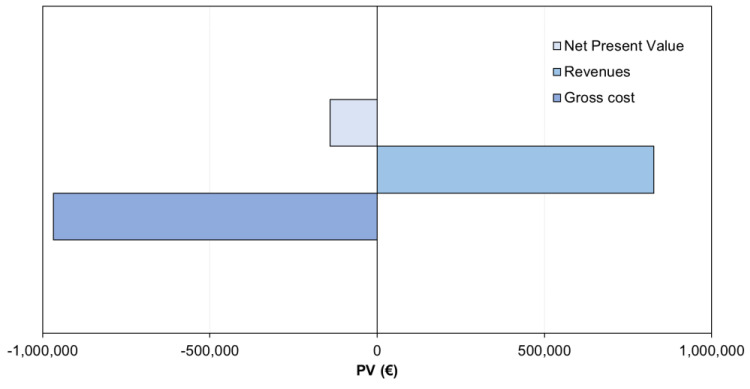
Gross cost, revenues and net present value for the nitrogen recovery scenario under study.

**Figure 7 membranes-12-01251-f007:**
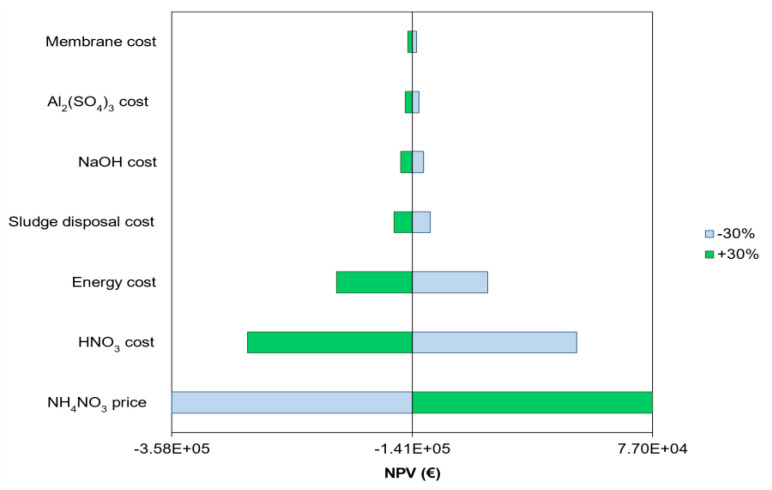
Sensitivity analysis for a ±30% variation of the main economic parameters.

**Figure 8 membranes-12-01251-f008:**
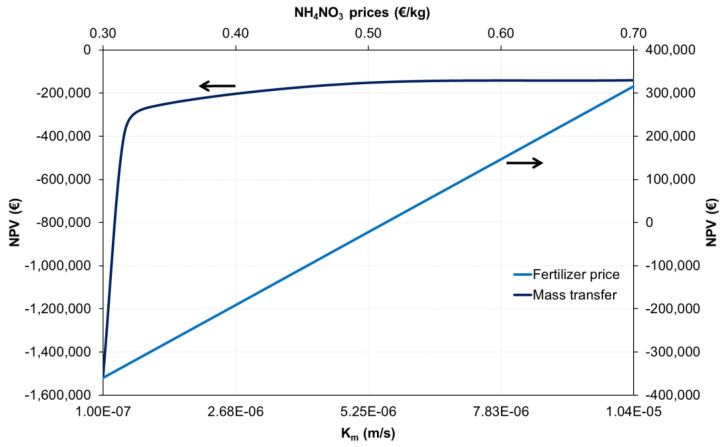
Sensitivity analysis for the NH_4_NO_3_ prices and mass transfer coefficient (K_m_). *The arrows indicated the *y*-axis corresponding to each line of the graph*.

**Table 1 membranes-12-01251-t001:** Most frequently used coagulants in water treatment according to bibliography.

N.º	Coagulant Used	Author
1	Tanfloc POP	[30]
2	Al_2_(SO_4_)_3_	[31]
3	FeCl_3_	[32]
4	FeCl_3_ + Clay Minerals	[33]
5	Lactic Acid	[34]
6	AlCl_3_	[34]

**Table 2 membranes-12-01251-t002:** Experimental conditions for optimal dosage determination.

Coagulant	Dosage (mg/L)	Mixing Time (min)	Mixing Speed (rpm)	Settling Time (min)
Optimal coagulant	10	5	200	30
	30			
	50			
	100			
	200			
	400			
	800			

**Table 3 membranes-12-01251-t003:** Individual dependent variables and their range of values.

Variable	Units	Studied Range
Mixing time (MT)	min	5; 15; 25
Mixing velocity (MV)	Rpm	100; 175; 250
Resting time (RT)	min	15; 30; 45

**Table 4 membranes-12-01251-t004:** Initial centrate characterization.

Parameter	Value	Unity
Sodium	474.4 ± 18.4	mg/L
TAN	650 ± 64.5	mg/L
Potassium	146.6 ± 7.6	mg/L
Magnesium	33.6 ± 13.4	mg/L
Calcium	90.5 ± 26.8	mg/L
Chlorine	348.0 ± 15.4	mg/L
Nitrate	30.7 ± 8.8	mg/L
Phosphate	138.1 ± 30.2	mg/L
Sulphate	37.5 ± 10.8	mg/L
Carbonates	3366.7 ± 792.5	mg/L
Turbidity	275.1 ± 106.2	NTU
COD	786.0 ± 126.7	mg O_2_/L
TSS	235.0 ± 104.7	mg/L
pH	8.2 ± 0.1	--

**Table 5 membranes-12-01251-t005:** Results obtained on COD removal (%) and turbidity reduction for the coagulation assay coagulant test.

Coagulant	COD Removal (%)	Turbidity Reduction (%)
Al_2_(SO_4_)_3_	50.2 ± 1.1	82.3 ± 1.1
Derypol HT20	36.0 ± 0.3	84.7 ± 0.4
FeCl_3_	38.9 ± 0.3	74.2 ± 1.7

**Table 6 membranes-12-01251-t006:** Results of water quality improvement for the coagulation experiments (COD removal (%), turbidity reduction (%)) as a function of coagulant type and coagulant dose.

Dosage	Al_2_(SO_4_)_3_			FeCl_3_		
(mg/L)	COD Removal	Turbidity Reduction	pH	COD	Turbidity Reduction	pH
10	51.5 ± 1.2	80.4 ± 2.8	8.0	42.5 ± 0.7	60.3 ± 1.2	8.0
30	56.2 ± 1.0	85.5 ± 4.4	7.7	48.0 ± 0.9	71.2 ± 1.2	7.9
50	50.1 ± 1.7	82.3 ± 3.5	7.4	38.9 ± 1.6	74.2 ± 2.4	7.7
100	41.1 ± 0.9	76.7 ± 1.2	7.1	41.5 ± 1.9	80.6 ± 3.4	7.4
200	62.1 ± 1.2	86.6 ± 4.0	6.9	45.1 ± 2.1	87.9 ± 3.3	7.1
400	66.7 ± 2.5	82.2 ± 1.7	6.1	50.0 ± 1.9	90.4 ± 4.1	6.7
600	64.7 ± 2.1	55.5 ± 2.4	4.3	52.5 ± 1.8	95.7 ± 3.4	6.4
800	66.9 ± 1.0	27.3 ± 3.3	4.1	51.8 ± 1.7	97.0 ±3.0	5.8

**Table 7 membranes-12-01251-t007:** Experimental removal using optimal conditions extracted from Design Expert 11. The errors represent standard deviation (n = 3).

Variables	Studied Conditions	Parameters	Experimental Removal (%)
Mixing time	5 min	COD	58.1 ± 0.3
Mixing velocity	100 rpm	TSS	94.9 ± 0.2
Settling time	30 min	Turbidity	89.8 ± 0.8

**Table 8 membranes-12-01251-t008:** Km values obtained in different studies with hollow fibre liquid–liquid membrane contactors.

Study	Mass Transfer (m/s)	Flow Rate (mL/min)	Type of Contactor	Initial [NH_3_] g/L	% Removal	Pre-Treatment	Water
This study	1.0 × 10^−5^	450 mL/min	FS-LLMC (PTFE)	0.9	66.6	C/F and Aeration	Sidestream
[14]	8.8 × 10^−7^	450 mL/min	HF-LLMC (PP)	3.9	76.1	Ion-exchange	Sidestream
[50]	8.9 × 10^−6^	920 mL/min	HF-LLMC (PVDF)	2–10	90.0	-	Synthetic
[15]	2.9 × 10^−7^	770 mL/min	HF-LLMC (PMP)	5.0	93.1	-	Synthetic
[51]	1.89 × 10^−6^	450 mL/min	HF-LLMC (PP)	1.7	85	Sorption	Sidestream

## Data Availability

Not applicable.

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
