# Peer review of "Impact of Sidestream Pre-Treatment on Ammonia Recovery by Membrane Contactors: Experimental and Economic Evaluation"

_membranes, 2022, doi:10.3390/membranes12121251_

Round 1

Reviewer 1 Report

See attached feedback file

Reviewer 2 Report

In the present paper, the authors conducted the techno-economic evaluation for ammonia recovery process through the combination of C/F, aeration, and membrane contactor (MC). The paper provided some new information for AD effluent concentrate pretreatment technologies. The experimental design can be reasonable. However, the introduction is not that scientific and logical. Furthermore, there are several points which must be improved and corrected further by the authors. Major revision is required prior to recommendation of its acceptance. My comments are as follows.

1.         Introduction: The novelty of paper may be the techno-economic evaluation for ammonia recovery by membrane contactor with pretreatment by C/F. However, in the line 90-92, the introduction of aeration is to reduce the amount of bicarbonates. What is the relationship between Aeration, C/F pretreatment and MC process for ammonia recovery? Notably, the combination of MC and C/F is discussed in the introduction. However, there is no related instructions for aeration. Hence, the discussion about this part needs to be strengthened. It is suggested to add a paragraph to explain this part.

2.         The initial concentrations of the wastewater weren’t given in chapter. The change of TSS was not mentioned when selecting flocculant. In practical application, will different concentrations of TSS affect the selection of flocculants?

3.         Note the superscript of "HCO3-" in the abstract.

4.         Figure 1 needs to be optimized. Also, please note the superscript of "NH4+" in the figures.

5.         Please add the error bar for COD removal in Figure 3. The bicarbonate mentioned in the title of Figure 3 is not shown in the figure, and the “COD (5)” in the title should be “COD (%)”?

6.         In the section 3.4, only the experimental results are simply described, and no in-depth analysis of the test results is performed.

7.         Line 438-440, “The higher Km achieved in this study could be attributed to the high efficiency of the pre-treatment process and the low duration of the experiments (16 h), leading to nearly negligible fouling during membrane contactor operation.” The potential membrane fouling can be negligible because of the short operation time (16 h)?

8.         line 448, “fouling” should include biological “biofouling” rather than juxtaposition.

9.         The number in Figure 5 is not accurate enough. Is the decimal point of the number in Figure 5 Misused as a comma?

10.     Revise the superscripts and subscripts of all the chemical formulas, such as Line 24, Link 47, Line 140, Line 332…

11.     Line 71-73: “such as UF [19] coagulation/flocculation (C/F) processes [17] or [20] that used ion exchange technology to reduce COD”. In this sentence, are “UF” and “C/F” the same technology? If not, you should write as “UF and coagulation/flocculation (C/F)”.

12.     Line 99: Please give the consistent expression on “C/F”. Herein is “coagulation-flocculation” but is “coagulation/flocculation” as stated before.

13.     Line 103: what is meaning of “IV”?Please give the definition of “IV”.

14.     Please check your writing about relation of juxtaposition, for example, it should be modified as “58.1± 0.3 COD, 94.9 ± 0.2 TSS, and 89.8 ± 0.8 turbidity” in the Line 366.

15.     Equation (4)-(6): Firstly, please check your manuscript all the accuracy of the chemical formula and equation. Secondly, Eq. (4)-(6) should be reversible reaction. Therefore, you should use the reversible symbol (). Thirdly, please add the state of chemical (such as NH+ 4(aq), NH3(g)) in these equation because you already give the gaseous CO2(g) and the aqueous CO2(aq) in the Eq.(4).

16.     The pictures in the paper can be further optimized to clearly illustrate your results.

17.     The language should be further polished to avoid the basic errors.

Reviewer 3 Report

This manuscript studies coagulation-flocculation and aeration pre-treatments to reduce chemical oxygen demand (COD), turbidity, suspended solids and alkalinity before the ammonia recovery stage using a membrane contactor, including a techno-economic evaluation. The obtained results are relevant for recovery of ammonia from anaerobic digestion effluents. However, before being suitable for publication, some changes would be needed in the manuscript. 

  • -Page 6, equation 6. Due to pH increase, ammonia could volatilise. Was this possible loss of ammonia controlled or quantified? 

  • -Page 10, Figure 3. The values for no flocculant addition, are different than the ones presented in Table 6. Could the authors explain why? 

  • -Page 11, Line 410. An evaluation of the amount of NaOH saved by aeration is this study would be useful. Was this parameter measured during the experiments? 

  • -Page 12, Figure 5b. Why the initial pH of no treated digestate is 9,12, while the treated one is 10,25. In page 6, line 230, it is stated “The pH of the feed solution was increased up to 10.2 with NaOH 1 M”. This pH increase was performed in treated and untreated digestate? If not, differences in pH may influence ammonia removal. 

  • -Page 12, Table 7. Authors should include a line in the table with the data of this study for better comparison. 

Other comments: 

  • -Page 2, Line 59, 72 and 74. Please, write the name of the author before the reference numbers [16], [20] and [21]. 

  • -Page 4, Line 140. Subscripts are missing in (Al2(SO4)3) and (FeCl3) 

  • -Page 6, eq (4). There is a plus symbol to spare before the H+: HCO3 - ++H +→ +CO2(aq)+H2O 

  • -Page 8, Line 321. “COD removal efficiency increased from 42.5 to 51.2%”. The second percentage do not match with the value in Table 5. The same in line 323, “the COD removal efficiency increased from 51.5 to 66.8%”. Please, correct. 

  • -Page 9, Line 332. Please, revise this sentence. 

  • -Page 10, Figure 3. Please, correct the word “floculant” by “flocculant”.

Round 2

Reviewer 2 Report

NO

Reviewer 3 Report

After the changes that have been made by the authors following the reviewers’ comments, the manuscript is now suitable for publication.